# E2 Partner Tunes the Ubiquitylation Specificity of Arkadia E3 Ubiquitin Ligase

**DOI:** 10.3390/cancers15041040

**Published:** 2023-02-07

**Authors:** Georgia N. Delegkou, Maria Birkou, Nefeli Fragkaki, Tamara Toro, Konstantinos D. Marousis, Vasso Episkopou, Georgios A. Spyroulias

**Affiliations:** 1Department of Pharmacy, University of Patras, 26504 Patras, Greece; 2Department of Brain Sciences, Imperial College, London W12 0NN, UK

**Keywords:** Arkadia, RING domain, E2 enzymes, ubiquitylation, NMR spectroscopy

## Abstract

**Simple Summary:**

Ubiquitin-mediated proteasomal degradation is a fundamental and tightly coordinated process that controls the cellular concentration of proteins. E3 ubiquitin ligases are responsible for substrate recognition, and their deregulation is associated with various diseases, most notably, cancer. The E3 ligase Arkadia (*RNF111*) targets for proteasomal degradation negative regulators of the TGF-*β* SMAD2/3 signaling pathway and poly-SUMOylated proteins, e.g., promyelocytic leukemia protein. Herein, we provide valuable insights into the interaction and function of Arkadia with its physiological E2 partners, UbcH5Β and UbcH13, as well as UbcH7. Specifically, we identify key amino acid residues on these E2 enzymes and Arkadia that are essential for their pairing and control of the enzymatic activity of Arkadia in ubiquitylation machinery. Our findings enrich the current knowledge on the pivotal components of the ubiquitylation system and can be the basis for future drug development.

**Abstract:**

Arkadia (*RNF111*) is a positive regulator of the TGF-*β* signaling that mediates the proteasome-dependent degradation of negative factors of the pathway. It is classified as an E3 ubiquitin ligase and a SUMO-targeted ubiquitin ligase (STUBL), implicated in various pathological conditions including cancer and fibrosis. The enzymatic (ligase) activity of Arkadia is located at its C-terminus and involves the RING domain. Notably, E3 ligases require E2 enzymes to perform ubiquitylation. However, little is known about the cooperation of Arkadia with various E2 enzymes and the type of ubiquitylation that they mediate. In the present work, we study the interaction of Arkadia with the E2 partners UbcH5B and UbcH13, as well as UbcH7. Through NMR spectroscopy, we found that the E2–Arkadia interaction surface is similar in all pairs examined. Nonetheless, the requirements and factors that determine an enzymatically active E2–Arkadia complex differ in each case. Furthermore, we revealed that the cooperation of Arkadia with different E2s results in either monoubiquitylation or polyubiquitin chain formation via K63, K48, or K11 linkages, which can determine the fate of the substrate and lead to distinct biological outcomes.

## 1. Introduction

Ubiquitylation is an important post-translational modification that is catalyzed by the concerted action of activating (E1), conjugating (E2), and ligating (E3) enzymes. The E1 enzyme activates and transfers ubiquitin (Ub) to the catalytic cysteine residue of the E2 enzyme to generate an E2~Ub thioester intermediate. Afterwards, the Ub moieties attach to a nucleophile, which is usually a lysine residue on the protein substrate, a process mediated by the E3 ligase [1]. This results in the conjugation of monoubiquitin or polyubiquitin chains of different linkages and lengths to substrates. Ubiquitylation has been found to be highly versatile, in terms of the range of cellular responses it can trigger. Several studies have demonstrated that it is involved in different aspects of cellular homeostasis, including receptor endocytosis, protein degradation, DNA repair, and apoptosis [2]. To accomplish such a broad spectrum of biological processes, a plethora of different proteins participates, such as two E1s, ~40 E2s, ~1000 E3s, and ~100 deubiquitinating enzymes (DUBs) [3]. Moreover, eight types of homogenous Ub chains (M1, K6, K11, K27, K29, K33, K48, and K63) and heterogeneous combinations of each linkage have evolved, determining the fate of the ubiquitylated substrate. Two types of polyubiquitin chains have been extensively studied so far. K48-linked polyubiquitin chain mostly targets the substrate for degradation via the ubiquitin-proteasome system (UPS), whereas K63 has a non-proteolytic role mainly in inflammation and DNA damage response [4].

Ubiquitin network deregulation can lead to many pathophysiological conditions, including cancer. Specifically, alterations in degradation via the UPS are observed in most cancer cells. Overexpression [5,6] and mutations [7,8] of various E3 ligases are involved in several cancers. Our expanding knowledge of the UPS and its role in human cancer has elicited great interest in the development of small molecules that target specific components of the pathway [9,10,11,12].

The E3 ubiquitin ligase Arkadia (*RNF111*) participates in several cellular functions and associated diseases. It was discovered as a positive regulator of the transforming growth factor-*β* (TGF-*β*) signaling pathway, essential for head development [13]. The substrates of Arkadia in the TGF-*β* pathway are the inhibitory SMAD7 and the transcriptional co-repressors SKI/SKIL when they are in complex with the receptor-activated effectors SMAD2/3 of the TGF-*β* pathway [14,15]. The enzymatic activity of Arkadia resides in the C-terminal RING-H2 domain (amino acids 942–983). Notably, Arkadia possesses three SUMO interaction motifs (SIMs) that enable Arkadia to recognize and target for degradation proteins modified by SUMO [16]. Therefore, Arkadia is one of the few SUMO-targeted ubiquitin ligases (STUbLs) identified so far and is implicated in the DNA damage response pathway. Specifically, Arkadia interacts with the UbcH13/MMS2 E2 complex to promote the formation of K63-linked chains to the SUMOylated xeroderma pigmentosum group C (XPC) protein, which flags damaged DNA. This results in the removal of XPC from damaged DNA and the enhancement of repair [17,18]. Furthermore, Arkadia is essential for the ubiquitylation and proteasome degradation of the SUMOylated promyelocytic leukemia protein (PML) in response to arsenic treatment [16]. Several studies link aberrant Arkadia function to cancer, including breast cancer [19], colorectal cancer [20,21], and non-small cell lung cancer [22]. As a critical regulator of TGF-*β*, Arkadia has a tumor-suppressive role in the early stages of cancer and a promoting role at the later stages of tumorigenesis including metastasis, i.e., tumor cell migration and invasion [23]. Despite the significance of Arkadia in many pathways, little is known about the features that modulate its enzymatic function.

A growing number of studies indicates that an E3 ligase interacts with more than one E2 [24,25]. Moreover, the selection of particular poly-Ub chain linkage appears to be determined by certain E2s (UbcH13/MMS2 [26], Ube2S [27], Ube2K [28], etc.) and the E2–E3 combinations involved. The E2–E3 partnership can be complex and requires meticulous study to shed light on how it affects the subsequent fate endowed onto the substrate. Enriching the knowledge of the E2–E3 interaction can help to increase the druggability of specific E2–E3 pairs and manipulate their pathogenic role in diseases including cancer [29]. It has been reported that Arkadia interacts with several E2s: UbcH6, UbcH9, Ubc16, the UbcH5 family [30], and UbcH13 [17]. Ubc16, Ubch5s, and UbcH13 consist only of the highly conserved ubiquitin-conjugating (UBC) domain, whereas UbcH6 and UbcH9 have an additional N-terminal extension. We recently showed that, unlike most other E3 ligases, Arkadia utilizes non-RING elements to stabilize the UbcH5B~Ub conjugate in the ‘closed conformation’, where the C-terminus tail of Ub is extended and the thioester bond is primed for nucleophilic attack [31]. Furthermore, the RING domain together with these non-RING elements (termed Ark LONG) are required for the formation of unanchored chains in cooperation with the E2 complex, UbcH13/MMS2 [31]. However, the requirement of the non-RING elements in the enzymatic function of Arkadia still remains unsettled. It is also unclear which are the protein segments that determine the binding specificity of Arkadia for these E2 enzymes, whether the direct physical interaction of an E2 with Arkadia forms an enzymatically active E2–E3 pair, and which are the features responsible for the stabilization of the active Arkadia-bound E2~Ub conjugates.

In the present study, we report that the interaction surface is consistent in all E2–Arkadia pairs examined, although the interaction affinity differs significantly depending on the specific E2 enzyme. We demonstrate that the L1 and L2 loops of E2 are crucial determinants and exhibit a central role in the formation of the Arkadia–E2 interface. Moreover, we show that Ub binding to the ‘backside’ of UbcH5B enhances Arkadia-mediated polyubiquitylation and that Arkadia itself possesses a ‘linchpin’ arginine, essential for the adoption of ‘closed conformation’. Additionally, using auto-ubiquitylation assays we reveal that the enzymatic activity of Arkadia with UbcH13 requires its non-RING elements, and this cooperative action leads to Ark LONG monoubiquitylation. In contrast, we demonstrate that the interaction of Arkadia with UbcH13/MMS2 or UbcH5B results in the formation of poly-Ub chains of different linkages. Collectively, the structural, biochemical, and functional findings of the current study shed light on the intricate enzymatic function of Arkadia.

## 2. Materials and Methods

### 2.1. Constructs, Protein Expression and Purification

The human Arkadia (*RNF111*) constructs used in this study include residues 927–994 (Ark RING 68 aa) and 876–994 (Ark LONG 119 aa) (Appendix A). These constructs were expressed and purified as described elsewhere [31]. Human UbcH5B (*UBE2D2*), UbcH13 (*UBE2N*) and MMS2 (*UBE2V2*) proteins were expressed using Rosetta™ 2(DE3) cells (Novagen, Darmstadt, Germany). Cells were grown at 37 °C in minimal medium supplemented with 1 g/L ^15^NH_4_Cl and 1 mL/L ^15^N BioExpress^®^ (Cambridge Isotope Laboratories). After OD_600_ reached values of 0.6–0.9, IPTG was added at a final concentration of 1 mM, and the temperature was lowered to 18 °C for UbcH5B and UbcH13 and 25 °C for MMS2. Cells were harvested after 16 h and were lysed by sonication. The proteins were purified by metal ion affinity chromatography using increasing concentrations of imidazole buffers (10, 20, 40 mM imidazole and 20 mM Na_2_HPO_4_, 500 mM NaCl pH 8). The His_6_-tag was cleaved after overnight incubation at room temperature with thrombin (Merck Millipore), and the proteins were eluted at 10 mM imidazole. This was followed by further purification with size exclusion chromatography by Superdex^®^ 75 10/300 GL column equilibrated in 50 mM K_2_HPO_4_ and 50 mM KH_2_PO_4_ pH 7. Human UbcH7 (*UBE2L3*) and ubiquitin were expressed and purified as described elsewhere [21,32]. Arkadia RING R983A, RING R983K, LONG R983A, LONG R983K, UbcH7 K96S, UbcH5B S22R, UbcH5B C85S, UbcH5B C85S-S22R, UbcH5B F62A, and all Ub mutants were generated by PCR-based QuickChange site-directed mutagenesis according to the manufacturer’s protocol (PfuUltra II Fusion High-fidelity DNA Polymerase, Agilent Technologies, Santa Clara, CA, USA). All the mutant sequences were verified by Sanger sequencing. The expression and purification protocols of the mutants were the same as the wild type (wt) proteins, except that for the expression of UbcH5B F62A, the culture temperature after induction was altered to 25 °C. Ubiquitin mutants with K48 only (K48O) and K11 only (K11O) were purchased from Boston Biochem (Minneapolis, MN, USA) and Sigma-Aldrich (St. Louis, MO, USA), respectively.

### 2.2. Amino acid Selective ^15^N Labeling and Reverse Labeling (Unlabeling)

The selective ^15^N labeling of UbcH5B F62A with ^15^N-Ala was achieved in the auxotrophic DL39(DE3) *E. coli* strain. As auxotrophic cells are not able to synthesize specific amino acids, the cells exclusively utilize the supplemented amino acids for protein synthesis. In total, 1 L culture of M9 medium was prepared containing 1 g NH_4_Cl, 5 g D-glucose, 1 mL ^14^N BioExpress^®^ (CIL), 400 mg ^15^N-Ala, 150 mg ^14^N-Phe, 90 mg ^14^N-Tyr, 400 mg ^14^N-Asp, 200 mg ^14^N-Ile, 200 mg ^14^N-Asn, 500 mg ^14^N-Gly, 100 mg ^14^N-Met, 210 mg ^14^N-Lys, 200 mg ^14^N-Leu, and 200 mg ^14^N-Val. The reverse labeling of ^15^N UbcH5B S22R was performed for arginines using Rosetta™ 2(DE3) cells (Novagen). This involved the selective unlabeling of all arginines in the protein, keeping the other amino acids uniformly ^15^N labeled. This was accomplished by supplying the *E. coli* cells with ^15^NH_4_Cl as the sole source of nitrogen, along with 0.2 g/L ^14^N-Arg [33].

### 2.3. Nuclear Magnetic Resonance (NMR) Titration

Arkadia polypeptides and E2 enzymes as well as UbcH5B and ubiquitin titration experiments were performed by recording ^1^H–^15^N HSQC spectra of labeled ^15^N polypeptide before and after the sequential addition of the unlabeled protein partner at 25 °C. The unlabeled protein was added in eleven steps (the number differed in some titrations) in order to reach the following ratios and eventually, the saturation of labeled/unlabeled protein: 1:0.25, 1:0.5, 1:0.75, 1:1, 1:1.25, 1:1.5, 1:1.75, 1:2, 1:2.25, 1:2.50, 1:2.75. Chemical shift perturbation (CSP) after titration was calculated for each amide signal using the following formula [34,35]: $\Delta \delta_{ppm}=\sqrt{{\left(\Delta \delta_{HN}\right)}^{2}+{\left(\frac{\Delta \delta_{Ν}}{5}\right)}^{2}}$. The CSPs were derived calculating a threshold value for each of the studied interactions. The threshold value was set using the standard deviation [34]. NMR spectra were recorded on a Bruker Avance III 700 MHz spectrometer equipped with a four-channel 5 mm cryogenically cooled TCI gradient probe. Protein samples were prepared in a mixed solvent of 90% H_2_0 (50 mM K_2_HPO4, 50 mM KH_2_PO4 pH 7), 10% D_2_0, and 0.25 mM DSS (4,4-dimethyl-4-silapentane-1-sulfonic acid) as the internal chemical shift standard. The NMR data were processed with Topspin 3.2 pl5 Software and analyzed using computer-aided resonance assignment (CARA) [36]. The CSP results were mapped onto the structures using the UCSF Chimera Software.

### 2.4. Circular Dichroism (CD) Spectroscopy

CD measurements were carried out with a Chirascan spectrometer (Applied Photophysics Ltd., Surrey, UK). Far-UV spectra (190–260 nm) were recorded at 25 °C at a bandwidth of 1 nm and an integration time of 2 s, in a 0.05 cm cuvette. Protein samples were prepared in 50 mM K_2_HPO_4_, 50 mM KH_2_PO_4_ pH 7, and 20–50 μM concentration. All spectra were buffer corrected, smoothed (window factor of 3, Savitzky–Golay method), and analyzed using Pro-Data Viewer. Data plots were generated using the software Chirascan (Applied Photophysics Ltd., Leatherhead, UK).

### 2.5. Isothermal Titration Calorimetry (ITC)

ITC experiments were performed using a MicroCal PEAQ ITC (Malvern, UK) at 25 °C. Proteins were dialyzed into a 50 mM KH_2_PO_4_ and 50 mM K_2_HPO_4_ pH 7 buffer. All samples were degassed. The sample cell contained 100–300 μM of Arkadia polypeptides, while 1–3 mM E2 was delivered by a series of 2 μL injections from the syringe with constant stirring (750 rpm) for a total of 15–19 injections. Each of the injections was separated by 210 s intervals. In each experiment, injections of protein into the buffer were conducted to remove the background signal. The raw data were processed in Microcal PEAQ ITC analysis software (Malvern) using a single-site model. Experiments were performed in duplicates.

### 2.6. Synthesis of E2-Ub Conjugate

For experiments using oxyester-linked UbcH5B–ubiquitin complexes, an active site Cys-to-Ser mutation (UbcH5B C85S) was performed. The resulting bond was only one atom different from the wt thioester and significantly more stable. A successful conjugation reaction was typically accomplished by mixing 1 µM E1, 150–200 µM E2, 500–600 µM ubiquitin, cycling buffer (50 mM Tris-HCl, 150 mM NaCl, pH 7), 10 mM creatine phosphate (Sigma-Aldrich), ~0.6 units mL^−1^ creatine kinase (Sigma-Aldrich), 5 mM MgCl_2_, and 5 mM Adenosine Triphosphate (ATP). Conjugation reactions were incubated at 37 °C for 16–20 h, and the completeness of the reaction was monitored using sodium dodecyl-sulfate polyacrylamide gel electrophoresis (SDS-PAGE). Oxyester-linked UbcH5B–ubiquitin was purified from E1 and unreacted proteins using a Superdex^®^ 75 10/300 GL size-exclusion chromatography column (GE Healthcare) equilibrated in 50 mM K_2_HPO_4_ and 50 mM KH_2_PO_4_ pH 7 buffer. The purified C85S UBCH5B–Ub conjugate was collected, flash-frozen in liquid nitrogen, and stored at −80  °C. The same protocol was performed in the synthesis of C85S-S22R UbcH5B-Ub conjugate.

### 2.7. Oxyester Hydrolysis Assays

Ε2–Ub conjugate hydrolysis assays were carried out by incubating 50 μM of E2–Ub oxyester complex with 25 μM of Arkadia polypeptides and 100 μM Ub (when appropriate) in 50 mM K_2_HPO_4_, 50 mM KH_2_PO_4_ pH 7, at 25 °C for 0.5, 1, 2, 3, and 4 h and, when necessary, 16 h. Reactions were terminated by the addition of SDS loading buffer. The protein samples were separated by 15% SDS-PAGE, and the gels were stained with Coomassie blue stain. The hydrolysis of oxyester-linked E2–Ub was quantified with Image Lab (ChemiDoc Imaging System, Biorad, Hercules, CA, USA).

### 2.8. Ubiquitylation Assays

In vitro ubiquitylation assays were performed by incubating 1 μM E1, 5 μM UbcH5B or UbcH7 or 10 μM UbcH13/MMS2, 15 μM Arkadia or its variants, and 150 μM Ub in 20 mM Tris-HCl, 50 mM NaCl pH 7.5, 5 mM ATP, 2 mM MgCl_2_, and 2 mM Dithiothreitol (DTT). The ubiquitylation protocol for the UbcH5B and UbcH7 mutants was the same as in wt proteins. When appropriate, ubiquitin variants were used (Ub K6R, Ub K11R, Ub K27R, Ub K29R, Ub K33R, Ub K48R, Ub K63R, Ub K48O, and Ub K11O, 150 μΜ). The reactions were incubated at 37 °C, and samples were collected after 0, 5, 10, 20, 30, and 60 min. Assays were stopped with SDS/DTT loading buffer, and the reaction products were separated by reducing 15% SDS-PAGE followed by Western blotting using an anti-Ub antibody (Santa Cruz Biotechnology, SCB, Dallas, TX, USA) and visualized using ChemiDoc Imaging System (Biorad). All images were taken at the same time of exposure.

## 3. Results

### 3.1. The Phe62 Residue of UbcH5B Is Critical for the Interaction with Arkadia

Here, we focused on the identification of the key factors and amino acids required for the formation of functional Arkadia–E2 complexes. Published data show that Arkadia (Ark) RING, primarily, interacts with UbcH5B through its *a*-helix and its two zinc-binding loops, whereas UbcH5B uses its *α_1_* and *α_2_*-helixes as well as its L1 and L2 loops to contact the RING domain [37]. Furthermore, the interaction of UbcH5B with the Ark LONG polypeptide (which includes the RING and the non-RING elements adjacent to its C-terminus [31]) results in stronger affinity than the RING domain, and this complex is active in the auto-ubiquitylation assay in vitro. UbcH5B amide resonances Ile6, Asp12, Phe62, Ser94, Ala96, Leu97, Thr98, Ile99, Ser100, Val102, and Leu103 disappeared and remained undetectable in the presence of Ark LONG [31]. Ile6 and Asp12 correspond to *a_1_*-helix, yet this region serves as a docking site for the E1 enzyme to form the E2~Ub conjugate. Thus, these amino acids cannot be targeted for mutational analysis. Phe62 and the group of Ser94, Ala96, and Leu97 reside in the L1 and L2 loops, respectively (Figure 1). The sequence alignment of different Class I E2 enzymes confirmed the conservation of Phe62, Ser94, Pro95, Ala96, and Leu97 in most UBC domains (Figure 1). Therefore, we speculated that those amino acids in the L1 and L2 loops play key roles in the formation of the E2–Arkadia interface.

We first assessed the role of Phe62 in the Arkadia–UbcH5B interaction. We mutated the Phe62 of UbcH5B to alanine (F62A) and tested the ability of the mutant to interact with the Ark RING and Ark LONG and perform auto-ubiquitylation. Firstly, we examined through CD and NMR spectroscopy the effect of this mutation on the UbcH5B secondary structure. The CD spectra of UbcH5B and UbcH5B F62A (Appendix A) were identical. Additionally, the overlay of the ^1^H–^15^N HSQC spectra of the two polypeptides indicated minor changes to the peak dispersion; therefore, significant conformational changes are not expected (Appendix A). The above data led to the conclusion that any alteration in UbcH5B F62A activity is not related to structural changes. The identification of the A62 resonance was achieved through the selective ^15^N labeling of UbcH5B F62A with ^15^N-Ala (Appendix A) to obtain the mutant assignment and proceed with NMR titration experiments.

Then, the interaction of the Ark RING with UbcH5B F62A was monitored through the titration of the ^15^N-labeled Arkadia sample with non-labeled UbcH5B F62A and *vice versa*. CSP data analysis illustrated the weaker interaction of the F62A E2 with the Ark RING polypeptide, compared with that of the wt E2 (Figure 2a,b and Appendix A). The CSPs of the RING residues were significantly smaller than those observed with the wt UbcH5B, reflected also at the threshold of each graph (0.07 and 0.12 ppm, respectively). Notably, Val969, Asp970, and Trp972 of the Ark RING exhibit a free-to-bound chemical exchange equilibrium in the ‘fast exchange’ timescale during the titration with UbcH5B F62A, in contrast to the titration with the wt UbcH5B, where their resonances disappeared and remained undetectable. These amino acids are in the *a*-helix of the RING domain which directly contacts the UbcH5B enzyme, indicating that UbcH5B F62A mutation alters the spatial topology of the crucial structural elements, disturbs the E2–E3 interface, and finally, attenuates the E2–E3 interaction. Further CSP analysis of the UbcH5B F62A residues after the addition of the Ark RING (Figure 2c) showed weaker interaction compared to the titration with the wt UbcH5B [31]. Additionally, the interaction of the UbcH5B F62A with the Ark LONG (Figure 2d), monitored by NMR, indicated a weaker affinity between the two polypeptides. More specifically, no signal broadening and disappearing were observed during both titrations, and their CSP graphs were identical. To confirm the decreased affinity between the UbcH5B F62A mutant and both Arkadia constructs, ITC experiments were carried out (Figure 2e,f).

The dissociation constant (*K*_d_) of the Ark RING with the F62A mutant was measured at 0.13 mM, whereas the *K*_d_ of the Ark RING with wt UbcH5B is 32 μM [31] (Table 1). Moreover, the *K*_d_ of the Ark LONG with F62A was measured at 0.19 mM compared to that of the Ark LONG with the wt UbcH5B that is 2.7 μM (Table 1). Consistent with the NMR data, the interaction of UbcH5B F62A with RING or LONG polypeptide displayed dissociation constants in agreement with each other. Furthermore, both interactions were endothermic and entropy driven. The above findings with the use of different approaches confirmed that the F62A mutant disables the formation of a canonical UbcH5B–Arkadia complex by weakening the interaction and the affinity between the two polypeptides.

The impact of the F62A mutation on the ubiquitylation machinery was investigated through an in vitro auto-ubiquitylation assay with the use of the Ark LONG polypeptide. We found that polyubiquitylation with the wt UbcH5B is achieved in 5 min, whereas, with the F62A mutant, it is achieved in 60 min (Figure 2g). Therefore, F62 residue on UbcH5B is required for an efficient interaction with Arkadia and its polyubiquitylation in vitro.

### 3.2. The SPA Motif Acts as a Specificity Determinant for the E2–Arkadia Interaction

Our following goal was to investigate the role of the Ser94, Pro95, and Ala96 (SPA) motifs of the L2 loop (Figure 1) in the Arkadia–UbcH5B interface formation. To accomplish that, we used the E2 enzyme UbcH7, which has 37% percent identity to the UbcH5B enzyme but contains a KPA motif instead of the SPA motif. UbcH7 is an E2 enzyme that exhibits activity with the homologous to the E6AP carboxy terminus (HECT) and the RING-in-between-RING (RBR) family of E3s [38]. UbcH7 interacts with E3 RING ligases but lacks the ability to function with them. CSP data analysis showed that UbcH7 contacts the Ark RING through its *α_1_* and *α_2_*-helixes as well as L1 and L2 loops (Figure 3c,d) like UbcH5B.

Nonetheless, the Ark RING mainly interacts with UbcH7 through its *a*-helix, suggesting a relatively limited interaction surface (Figure 3a,b). Specifically, only the residue V969 of Arkadia exhibited ‘slow exchange’ free-to-bound equilibrium on the NMR chemical shift timescale. Thus, a ubiquitylation assay to evaluate the functionality of UbcH7 was performed. UbcH7 was unable to perform in vitro ubiquitylation with Arkadia (Figure 3h). ITC data demonstrated that the Ark RING-UbcH7 and Ark LONG-UbcH7 (Figure 3f and Appendix A) complex affinities are approximately equal to those of Ark RING–UbcH5B and Ark LONG–UbcH5B (27.3 μM, 4.3 μΜ and 32 μM, 2.7 μΜ, respectively) (Table 1). Notably, the non-RING elements of Arkadia increased the affinity of the Ark LONG for UbcH7.

Sequence alignment of E2 enzymes (Figure 1) shows that UbcH7 possesses a K96-P97-A98 amino acid segment instead of the SPA motif. Therefore, we introduced the K96S mutation to the UbcH7 enzyme to embed the conserved SPA motif on this E2 enzyme. The CD spectra of UbcH7 and UbcH7 K96S (Appendix A) were identical, whereas the overlay of their ^1^H–^15^N HSQC spectra indicated minor changes to the peak dispersion (Appendix A).

Strikingly, the K96S mutation caused a twofold increase in the threshold value of the UbcH7 K96S–Ark RING CSP graph (0.023 and 0.048 ppm, respectively) (Figure 3e). More importantly, the large number of CSPs above the threshold suggested a more extended interaction surface between the UbcH7 K96S–Ark RING compared to that of the wt UbcH7–Ark RING. ITC data further confirmed the enhancement of the E2–E3 interaction caused by K96S mutation, as the measured dissociation constant between the Ark RING and the UbcH7 K96S was 10.2 μM (Figure 3g). It is also important to underline that the Ark RING–UbcH7 K96S interaction displayed different thermodynamic parameters to the wt complex. The Ark RING–UbcH7 interaction is endothermic and entropy-driven, whereas the Ark–UbcH7 K96S interaction is entropy- and enthalpy-favorable. The above data suggest that the mutant UbcH7 exhibits a higher affinity to Ark RING, establishing a stronger interaction compared to the native E2 enzyme. However, despite the noteworthy enhancement in E2–E3 affinity attributable to K96S mutation, the mutant failed to form an enzymatically active E2–E3 pair with the Ark LONG and promote auto-ubiquitylation (Figure 3h). Previous studies have reported that UbcH7 fails to function with E3 RING ligases in vitro [38]. Specifically, UbcH7 exhibits reactivity solely with cysteine (transthiolation reaction) and not with lysine as a Ub receptor, making it a HECT/RBR-only E2 enzyme [38,39]. Although the Ark LONG construct possesses two extra non-Zn^2+^-liganding cysteines (C905 and C926 in Appendix A), neither UbcH7 nor UbcH7 K96S could transfer Ub cargo to Ark LONG, implying a different mechanism of function.

### 3.3. Ub^B^-UbcH5B Binding Enhances Ark LONG-Mediated Ub Transfer

According to the literature, UbcH5B~Ub conjugate stabilization in the ‘closed conformation’ is enhanced by the non-covalent binding of Ub (Ub^B^) to the ‘backside’(opposite to its active site) region of UbcH5B [40,41]. To further investigate whether Ark LONG-induced ubiquitylation utilizes the Ub^B^ binding on the ‘backside’ region of UbcH5B, the UbcH5B S22R mutation was generated. The sequence alignment of UbcH5s and UbcH7 (which does not bind free Ub^B^) reveals that UbcH7 has an Arg at the position corresponding to UbcH5s Ser22 (Figure 1). Our CD and NMR data showed that the S22R mutation affected neither the UbcH5B secondary structure nor the UbcH5B–Ark RING interaction (Appendix A and Figure 4a,b).

To assign the HN resonance of the mutant, the reverse labeling of ^15^N UbcH5B S22R was performed for arginines (Appendix A). Afterwards, the interaction of Ub with wt UbcH5B or UbcH5B S22R was monitored through NMR-driven titrations (Figure 4d,e,g,h). Ser22 and Gly27, located in the *β*-sheet of UbcH5B, exhibited tight binding and a ‘slow chemical exchange’ equilibrium in the NMR timescale during the Ub^B^ interaction. Additionally, residues corresponding to the C-terminus of the *a_1_*- and *a_4_*-helixes, as well as the *β*-sheet, comprising the ‘backside’ region of UbcH5B, exhibited large CSPs (Figure 4f). The ubiquitin interaction surface also appeared to be well defined, consisting of amino acids on the C-terminus and an Ile44 hydrophobic patch of Ub (Figure 4i). Notably, Ala46 and Lys48 residues of Ub broadened beyond detection. The introduction of the S22R mutation abolished the interaction with Ub (Figure 4d,g). A comparison of the chain-building capacity of UbcH5B and UbcH5B S22R in collaboration with Ark LONG demonstrated that the lack of Ub^B^-UbcH5B binding slows the ubiquitylation rate. Specifically, the wt UbcH5B achieves in 5 min the same level of ubiquitylation as the S22R achieves in 20 min (Figure 4j). The Ub transfer by Ark LONG in the presence or absence of Ub^B^ was further estimated by measuring the ability of Arkadia to promote the hydrolysis of the C85S UbcH5B-Ub and the C85S-S22R UbcH5B-Ub conjugate, respectively. The disruption of Ub^B^ binding reduced the rate of conjugate hydrolysis over time (Figure 4k). The overall conclusion from the above experiments is that the non-covalent binding of Ub^B^ to the ‘backside’ region of UbcH5B enhances the ubiquitylation mediated by Ark LONG. A previous study suggests that the formation of higher-order E2~Ub complexes through Ub^B^ binding enhances the hydrolysis of the E2~Ub complex and thus the ubiquitination process [42]. Furthermore, it is reported that Ub^B^ acts as an allosteric regulator that increases the E3-E2~Ub affinity and promotes Ub transfer [40].

### 3.4. Arkadia Possesses a ‘Linchpin’ Arginine

Several RING E3s, including Ark2C [43], possess a conserved H-bond donor residue (typically arginine, lysine, or asparagine), which contributes to the shift in E2~Ub conformational equilibria toward a ‘closed conformation’, serving as a so-called ‘linchpin’ [44,45,46,47]. By sequence homology, we found that Arkadia possesses a positively charged residue near the C-terminal end of its RING core, Arg983 (Appendix A). Thus, we suggested that Arkadia could facilitate ubiquitylation allosterically via this ‘linchpin’ arginine. R983A mutation was designed and introduced to the Ark RING and Ark LONG constructs. Overlaying the ^1^H–^15^N HSQC spectra of the Ark RING R983A and the wt Ark RING indicated minor changes to the peak dispersion (Appendix A); therefore, no significant conformational changes were imposed by the replacement of Arg983 by alanine between the two polypeptides.

Additionally, CSPs showed that the Ark RING R983A interacts with UbcH5B in a way similar to the wt polypeptide [31] (Figure 5a,b and Appendix A). Specifically, the dissociation constant of the Ark LONG R983A–UbcH5B complex was measured at 8.4 μΜ compared to 2.7 μΜ of wt Ark LONG–UbcH5B (Table 1 and Appendix A). Notwithstanding, we observed a 20 min delay in the ubiquitylation of the Ark LONG R983A mutant (Figure 5e). The mutant also inhibited the oxyester hydrolysis (Figure 5f) as the remaining complex concentration was almost equal to the initial one (43 and 50 μM, respectively). These findings support that Arkadia-mediated E2~Ub stabilization and Ub transfer depend on the presence of the ‘linchpin’ residue and not simply on the E2–E3 complex formation.

To further investigate the significance of the R983 amino acid in the mechanism of action of Arkadia, a conservative arginine-to-lysine substitution was performed. Lysine is a common amino acid that plays the role of ‘linchpin’ residue in some E3 ligases instead of arginine, e.g., Breast Cancer type 1 (BRCA1) [48]. In agreement with R983A, our NMR and ITC data demonstrated that R983K mutation did not affect the Ark RING secondary structure and the interaction between UbcH5B–Ark RING R983K and UbcH5B–Ark LONG R983K (Figure 5c,d and Appendix A). In particular, the dissociation constant of the Ark LONG R983K and UbcH5B was measured at 5.5 μM, approximately equal to the one of Ark LONG-UbcH5B (Table 1 and Appendix A). However, the introduction of R983K mutation to Arkadia disabled both ubiquitylation and oxyester hydrolysis assays (Figure 5e,f), even though arginine and lysine have similar properties and CSPs displayed no differentiation in the E2–E3 interaction. Altogether, the aforementioned results show that an R983 linchpin-connected E2–Ub–E3 complex is essential for the ubiquitylation activity of Ark LONG.

### 3.5. UbcH13 Interacts with Arkadia with Weaker Affinity than UbcH5B and Leads to Its Monoubiquitylation

Our next goal was to assess the interaction interface between Arkadia and the UbcH13 E2 enzyme. This enzyme recruits the E2 variant MMS2 and forms a complex that serves as a physiological partner for Arkadia. The interaction of Ark RING with UbcH13 was monitored through the titration of a ^15^N-labeled Arkadia sample with non-labeled UbcH13 and vice versa. CSP analysis indicated that Ark RING mainly interacts with UbcH13 through its *a*-helix and its two zinc-binding loops (Figure 6b). Specifically, residues Lys941, Gln966, Val969, Asp970, Lys978, and Glu987 of Arkadia exhibited a ‘slow exchange’ equilibrium on the NMR chemical shift timescale and remained undetected (Figure 6a). In agreement with the UbcH5B enzyme, UbcH13 uses its *α_1_*- and *α_2_*-helixes as well as L1 and L2 loops to contact the RING domain (Figure 6c,d). The resonance peak of Ser96 residue was the only one that disappeared during the titration. Ser96 belongs to the UbcH13 SPA motif indicating the central role of this motif in the interaction of different E2s with Arkadia. The NMR interaction of UbcH13 with the LONG construct displayed a more extended surface of interaction consisting of the *α_1_*- and *α_2_*-helixes, the L1 and L2 loops, and the *β*-sheet (Figure 6e,f). Notably, in addition to Ser96, residues Ile8, Lys10, Tyr76, Leu99, and Gln100 of UbcH13 disappeared during the titration. Consistent with the NMR data, the UbcH13–Ark RING dissociation constant compared to that of UbcH13–Ark LONG (264 and 51.9 μΜ, respectively) was lower by fivefold, as probed by the analysis of the thermodynamic parameters of the interaction through the ITC method (Figure 6g,h), proving the pivotal role of the non-RING elements of Arkadia in the binding affinity for UbcH13. Furthermore, it is important to point out the 19-fold increase in the *K*_d_ values of UbcH13–Arkadia compared to those measured for UbcH5B–Arkadia (Table 1), suggesting weaker binding between Arkadia and the UbcH13 enzyme.

We also tested the cooperation of UbcH13 with Ark RING and Ark LONG in auto-ubiquitylation assays, and we found that UbcH13–Ark LONG synergistic action resulted in the monoubiquitylation of Ark LONG, whereas Ark RING was not capable of performing monoubiquitylation in the presence of UbcH13 (Figure 6i). To separate the monoubiquitylated band from the UbcH13-Ub band, we repeated the assay using GST-tagged Ark LONG and Ark RING polypeptides in ubiquitylation with UbcH13. A single band with the anticipated molecular weight (~49 kDa) for the complex GST-Ark LONG-Ub was present on the Western blot (red box in Figure 6i), confirming the ability of UbcH13 to function only with Ark LONG and result in its monoubiquitylation.

### 3.6. Ark LONG Enhances the Ability of UbcH13/MMS2 to form Free Ub Chains Using ‘Linchpin’ R983 of Arkadia

We have previously demonstrated that Ark LONG enhances the formation of unanchored K63-linked polyubiquitin chains in synergy with the UbcH13/MMS2 complex [31]. Specifically, the presence of Ark LONG in the ubiquitylation assays increased the rate of free chain formation by 20 min compared to UbcH13/MMS2 alone (Figure 7a). We then examined whether the R983 of Ark LONG is required for the enhancement of unanchored Ub chain formation by the MMS2/UbcH13 complex. We found that Ark LONG R983A and Ark LONG R983K delayed the initiation of ubiquitin chain assembly compared to wt by 5–10 min (Figure 7a,b), suggesting that the ‘linchpin’ arginine of Arkadia is implicated also in the Arkadia–MMS2/UbcH13~Ub stabilization.

### 3.7. UbcH5B Promotes K48- and K11-Linked Polyubiquitylation of Arkadia

Our final goal was to determine the ubiquitin linkage type formed by UbcH5B–Arkadia synergistic action. To accomplish that, each of the seven lysines of ubiquitin was individually changed to an arginine. Ubiquitylation reactions were performed with wt Ub and with each of the seven Ub mutants (Figure 8a,b and Appendix A). Only the K48R and K11R Ub mutants displayed delayed initiation of ubiquitylation by 10 min compared to the wt. We assumed that the UbcH5B–Arkadia pair mediates the formation of either K48- or K11-linked chains. Indeed, ubiquitylation reactions performed with the use of K48O or K11O Ub mutants, which have all lysines mutated except K48 and K11, respectively, showed no significant difference compared to the wt Ub (Figure 8c). Both K48- and K11-linked chains have well-established roles in triggering proteasome-mediated degradation [27], suggesting that UbcH5B provokes the proteasomal degradation on the substrates of Arkadia.

## 4. Discussion

In the present study, we examined in depth the E2–Ark RING and E2–Ark LONG interactions through NMR spectroscopy, which provides insights into the dynamic nature of these crucial interactions. Through NMR titration experiments and interaction surface mapping, we studied the mechanism of E2 recruitment and the pivotal residues that contribute to the E3–E2 interaction and efficiency. We also identified through auto-ubiquitylation assays the lysine of Ub which is used for the ubiquitin chain formation and the fate of the ubiquitylated substrate in each E2–Ark LONG partnership. This knowledge provides accurate information to unsettled questions in the field of E3 ubiquitin ligases involved in cancer and other diseases.

Herein, we demonstrated that the non-RING elements of Ark LONG are critical for its interaction with the E2 enzymes UbcH13 and UbcH7, similar to what we have previously shown for the UbcH5B enzyme [31]. Additionally, the current study showed that Ark RING uses its *a*-helix and two zinc-binding loops to contact not only the UbcH5B enzyme but also UbcH13; both of them are well-established physiological partners of Arkadia. On the contrary, UbcH7 is an E2 enzyme that has not been shown to be functional with E3 RING ligases [38]. The NMR data suggest that the interaction surface between UbcH7 and the Ark RING is limited (only the *a*-helix of Arkadia is involved) and this can explain its incapability to collaborate efficiently with Arkadia in performing Ub transfer. In general, all of these three E2 enzymes utilize the same surface of their UBC domain to dock on the Ark RING. According to the NMR study presented here, the largest CSPs are located in the *α_1_*- and *α_2_*-helixes and the L1 and L2 loops, which are part of the canonical E2–E3 RING binding interface, explaining the extensive sequence conservation at these regions between the three E2 enzymes (Figure 9a).

However, the exact orientation of Arkadia bound to an E2 enzyme has yet to be determined. Superimposing the Ark RING structure (PDBid: 2KIZ) to the X-ray structure of the UbcH5B-c-CBL complex (PDBid: 4A49) resulted in a model of the Ark RING–UbcH5B complex (Figure 9c). A close-up view of the Ark RING–UbcH5B interface suggests that Pro61, Phe62, Ser94, Pro95, and Ala96 residues of UbcH5B directly contact the *a*-helix and the two zinc-binding loops of Arkadia (Figure 9b). Specifically, Phe62 seems to play a central role in the interface formation in agreement with our experimental results. According to the literature, UbcH5A Phe62 cooperates with the aliphatic residues Pro61, Pro95, and Ala96 and forms a hydrophobic interaction area against the helix and the two loops of Carboxy terminus of the Hsp70-interacting protein (CHIP) ligase [24]. In accordance with that, previous studies have established the importance of the hydrophobic properties of a specific tryptophan in RING domains (Trp972 in the *α*-helix of Ark RING [49]) in E2 recognition and binding [50]. The indole ring of Trp972 is packed into a mainly hydrophobic pocket and is oriented toward the solvent, probably contacting the hydrophobic area of the E2 enzyme, as depicted in the computational model (Figure 9b). UbcH13 contains a methionine (Met64) at the position equivalent to the Phe62 of UbcH5B. Met64 exhibited a large CSP in the titration with Ark RING (Figure 6c), indicating its important role in the UbcH13–Ark RING interaction. UbcH13 M64A mutant also abolishes the interaction with E3 ligases TNF receptor-associated factor 6 (TRAF6) and RNF8 [51,52] similarly to UbcH5B F62A that highly diminishes its interaction with the Rma1 RING ligase [42]. Altogether, these results support that E2–Arkadia complex formation involves extensive hydrophobic interactions and that the presence of a hydrophobic amino acid at a specific position in the L1 loop of E2 is necessary.

Moreover, the SPA motif appears to be a prerequisite for Arkadia recognition and efficient interaction with UbcH5B and UbcH13. The strongest evidence we provide in support of this claim is the introduction of the SPA motif to the UbcH7 enzyme, which lacks this specific amino acid segment. This mutation substantially enhanced the interaction of the mutant UbcH7 with Arkadia and imposed a significant shift in the thermodynamic parameters that characterize the interactions. Nevertheless, the presence of the SPA motif alone was not sufficient for promoting ubiquitin ligase function with Arkadia, consistent with previous studies on the CHIP ligase [53]. Apparently, the Ub transfer from the E2 enzyme to the acceptor E3 enzyme or the substrate necessary to complete the ligation, is based on the reactivity towards a lysine or cysteine, aside from the formation of a proper E2–E3 interface.

Ub^B^ binding is another major feature that differentiates those three enzymes in terms of their mechanism of action. UbcH7 and UbcH13 possess a positively charged residue at the center of their backside region (UbcH7 Arg23 and UbcH13 Lys24 in Figure 1), which prevents the interaction with Ub in this region. On the contrary, UbcH5B furnishes a platform to favor the non-covalent binding of Ub^B^. We demonstrated that this Ub serves as an allosteric activator of the Arkadia–UbcH5B~Ub complex to promote Ub transfer. Earlier studies demonstrated that a loss of Ub^B^ binding results in either monoubiquitylation instead of polyubiquitylation [54] or a significant decrease in polyubiquitylation [40]. A loss of Ub^B^ binding in the Arkadia–UbcH5B~Ub complex slows the rate of polyubiquitylation without inhibiting the process. Although the role of this non-covalent interaction in a cellular setting remains unclear, an increasing number of studies suggests that it is a delicate way to regulate the function of several E3 ligases.

With the aim of unraveling extra features involved in the Arkadia–E2~Ub complex stabilization, we studied the ‘linchpin’ Arg983 of Arkadia. This arginine resides right at the beginning of the C-terminal loop of the Ark RING (Figure 9d and Appendix A) and cements the E2 and Ub in the ‘closed’ active state through hydrogen bonding. The pivotal role played by the E3 hydrogen bond donor residue in ubiquitylation has been elucidated in several RING ligases, e.g., RNF4 and Ark2C [45,55]. Their ‘linchpin’ arginine contacts Gln92 of UbcH5 as well as Arg72 and Gln40 of Ub to shift the population of the UbcH5~Ub conjugate toward the ‘closed state’. Our results of Arkadia R983A mutant, which hindered ubiquitylation, are in line with these findings. The Arkadia R983K mutant also reduced ubiquitylation, albeit to a lesser degree. It has been recently reported that the presence of lysine instead of arginine at this position in several ligases, including BRCA1 (RNF53 in Appendix A), results in diminished ubiquitin transfer, revealing that those proteins have evolved attenuated activity [48]. Additionally, the present work enriches the knowledge of the ‘linchpin’-mediated Ub transfer. The interaction of the R983A and R983K Arkadia mutants with the UbcH13/MMS2 complex slowed down the Ub chain synthesis rate. All the above results provide convincing evidence that although the ‘linchpin’ arginine is required for efficient Ub chain assembly, it does not determine the specificity of Arkadia for the E2 enzymes. However, whether the residues of UbcH5B and UbcH13 involved in hydrogen bonding with R983 of Arkadia are equivalent needs further investigation.

Furthermore, we shed light on the Ub chain linkage achieved by Arkadia-mediated ubiquitylation. In the case of E3 RING ligases, the specification of chain linkage depends on the E2 that pairs with the RING ligase, as well as the substrate itself. Herein, we demonstrated that Arkadia bound to different E2 enzymes leads to different types of Ub modifications, and thus to different cellular outcomes. It is also important to highlight our finding that the non-RING elements of Arkadia are essential in order to perform ubiquitylation not only with UbcH5B and UbcH13/MMS2 [31], but also with the UbcH13 enzyme. Specifically, we showed that Arkadia builds polyubiquitin chains that mediate degradation, i.e., K48- or K11-linked, with UbcH5B, which is consistent with its ability to target for degradation substrates such as SMAD7, SKI, and SKIL. Additionally, the capacity of Arkadia to perform auto-ubiquitylation via UbcH5B and to form K48- or K11-linked polyubiquitin chains suggests that Arkadia regulates its own cellular levels. On the contrary, the interaction of Arkadia with UbcH13 alone leads to its monoubiquitylation, as previously reported for RNF4 ligase [56], whereas the cooperation of Arkadia with the UbcH13/MMS2 complex results in enhanced unanchored K63-linked polyubiquitin chain formation. Monoubiquitylation is so far implicated in protein subcellular localization, histone modification, viral budding, and DNA repair [57], whereas K63 linkage is involved in protein–protein interaction and DNA repair [58]. Hence, as both UbcH13 and the UbcH13/MMS2 complex collaborate with Arkadia and synthesize different types of ubiquitylation products, it is possible that their interplay controls the subcellular localization of Arkadia and the fate of its substrates. Nonetheless, further biological studies need to be carried out with Arkadia and its substrates in vivo.

## 5. Conclusions

Increasing interest in various aspects of drug discovery in the ubiquitylation system has developed a need for understanding the mechanism of the ubiquitylation components. So far, significant progress has been achieved in deciphering how RING E3s function. Arkadia is an example of a RING ligase with unique mechanism of action, as its RING domain is not fully active and requires non-RING elements to activate an E2~Ub complex. This study has the potential to provide an understanding of how Arkadia recognizes its physiological E2 partners and leads to the diverse linkage specificity of the polyubiquitin chain.

In that context, we demonstrate that the Arkadia–UbcH5B and Arkadia–UbcH13 complex interfaces remain constant in both cases. In contrast, the Arkadia–UbcH7 complex displays a relatively limited interaction surface in agreement with its inability to perform ubiquitylation. Although the E2–E3 complex interface possesses numerous interacting residues, a few hotspot residues significantly contribute to the molecular recognition process and the binding affinity between the E2–E3 partners. Based on the data presented in this study, the identity of residues at specific positions in E2 L1 and L2 loops and in the RING domain can tune the E2–Arkadia compatibility and result in functional E2–Arkadia pairs. Strikingly, Ub^B^ binding to UbcH5B also enhances Arkadia-mediated ubiquitylation. In conclusion, the collaboration of Arkadia with UbcH5B, UbcH13, and UbcH13/MM2 E2s results in different types of ubiquitylated products confirming the involvement of Arkadia in a broad spectrum of biological processes and highlighting the pivotal role of this E3 ligase in eukaryotic cells.

## Figures and Tables

**Figure 1 cancers-15-01040-f001:**
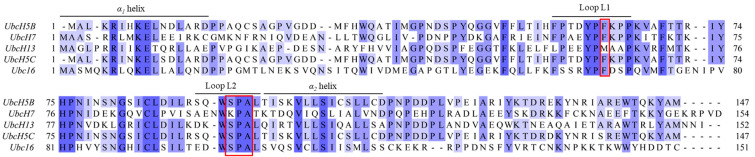
Sequence alignment of the UBC domain of different E2 enzymes produced by Jalview. Residues are colored according to the percentage of the residues in each column that agree with the sequence of UbcH5B (>80% dark blue, >60% medium dark blue, >40% light blue, ≤40% no color). The conserved phenylalanine and SPA motif at the L1 and L2 loops, respectively, are highlighted with red boxes.

**Figure 2 cancers-15-01040-f002:**
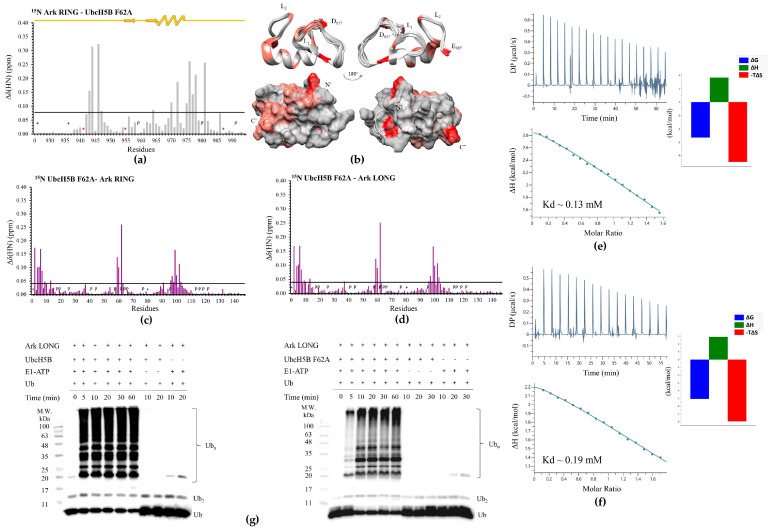
Interaction of E3 Arkadia with the E2 UbcH5B F62A mutant. (**a**) Diagram of the total CSPs measured at a 1:2 molar ratio (saturation point) for ^15^N Ark RING/^14^N UbcH5B F62A, *: represents disappeared residues, +: represents residues with no information. (**b**) Ark RING mapping (PDBid: 2KIZ) after addition of UbcH5B F62A (residues that disappeared during the interaction are colored red, whereas residues that exhibited ‘fast exchange’ interactions are colored coral). (**c**) Diagram of the total CSPs measured at a 1:2 molar ratio for ^15^N UbcH5B F62A/^14^N Ark RING. (**d**) Diagram of the total CSPs measured at a 1:2 M ratio for ^15^N UbcH5B F62A/^14^N Ark LONG. (**e**) ITC data for the titration of UbcH5B F62A into Ark RING and (**f**) into Ark LONG. (**g**) In vitro auto-ubiquitylation assays of Ark LONG using UbcH5B F62A (**right**) and wt UbcH5B (**left**).

**Figure 3 cancers-15-01040-f003:**
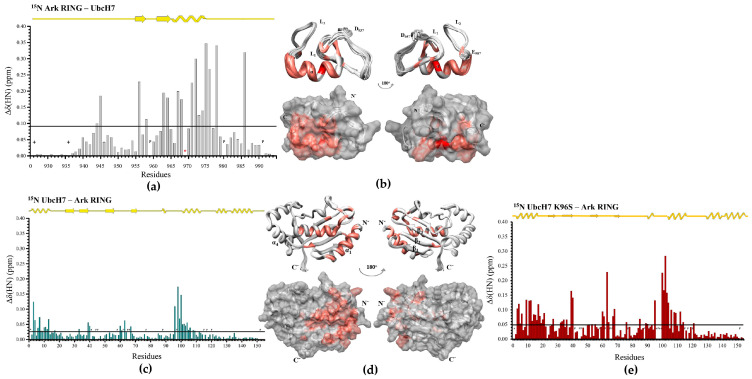
Interaction of E3 Arkadia with wt UbcH7 and UbcH7 K96S mutant. (**a**) Diagram of the total CSPs measured at a 1:2 molar ratio (saturation point) for ^15^N Ark RING/^14^N wt UbcH7, *: represents disappeared residues, +: represents residues with no information. (**b**) Ark RING mapping (PDBid: 2KIZ) after the addition of UbcH7 (residues that disappeared during the interaction are colored red, whereas residues that exhibited ‘fast exchange’ interaction are colored coral). (**c**) Diagram of the total CSPs measured at a 1:1.5 molar ratio for ^15^N UbcH7/^14^N Ark RING. (**d**) UbcH7 mapping (PDBid: 6XXU) after the addition of Ark RING. (**e**) Diagram of the total CSPs measured at a 1:1.5 molar ratio for ^15^N UbcH7 K96S/^14^N Ark RING. (**f**) ITC data for the titration of wt UbcH7 into Ark RING and (**g**) UbcH7 K96S into Ark RING. (**h**) In vitro auto-ubiquitylation assays of Ark LONG using wt UbcH7 and UbcH7 K96S, respectively.

**Figure 4 cancers-15-01040-f004:**
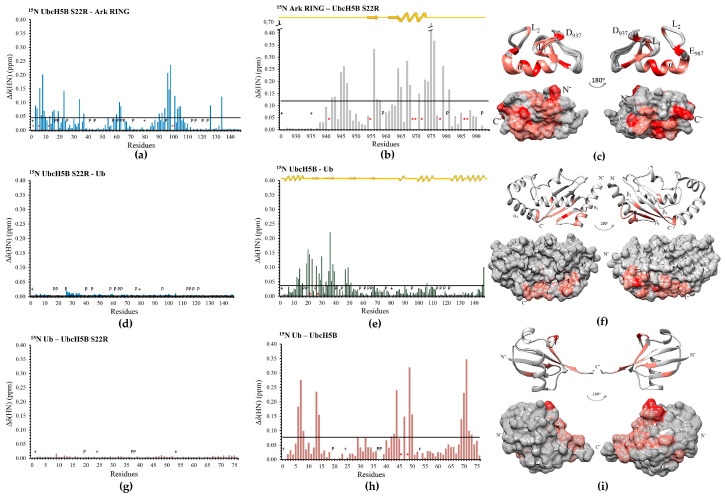
Implication of the Ub^B^ binding in Arkadia-mediated ubiquitylation. (**a**) Diagram of the total CSPs measured at a 1:2 molar ratio for ^15^N UbcH5B S22R/^14^N Ark RING, *: represents disappeared residues, +: represents residues with no information. (**b**) Diagram of the total CSPs measured at q 1:2 molar ratio for ^15^N Ark RING/^14^N UbcH5B S22R. (**c**) Ark RING mapping (PDBid: 2KIZ) after the addition of UbcH5B S22R (residues that disappeared during the interaction are colored red, whereas residues that exhibited ‘fast exchange’ interaction are colored coral). (**d**) Diagram of the total CSPs measured at a 1:2.75 molar ratio for ^15^N UbcH5B S22R/^14^N Ub. (**e**) Diagram of the total CSPs measured at a 1:2.75 molar ratio for ^15^N wt UbcH5B/^14^N Ub. (**f**) UbcH5B mapping (PDBid: 2ESK) after the addition of Ub. (**g**) Diagram of the total CSPs measured at a 1:2.25 molar ratio for ^15^N Ub/^14^N UbcH5B S22R. (**h**) Diagram of the total CSPs measured at a 1:2.25 molar ratio for ^15^N Ub/^14^N wt UbcH5B. (**i**) Ub mapping (PDBid: 1D3Z) after the addition of wt UbcH5B. (**j**) In vitro auto-ubiquitylation assays of Ark LONG using UbcH5B S22R (**left**) and wt UbcH5B (**right**). (**k**) (**Top**), oxyester hydrolysis assays showing the disappearance of C85S UbcH5B-Ub and C85S S22R UbcH5B-Ub in the presence of Ark LONG, over time. (**Bottom**), densitometry quantification from gels (bars indicate the range of experimental duplicates, * indicates statistically significant difference, *p* value ≤ 0.05). The uncropped WB images can be seen in Appendix A.

**Figure 5 cancers-15-01040-f005:**
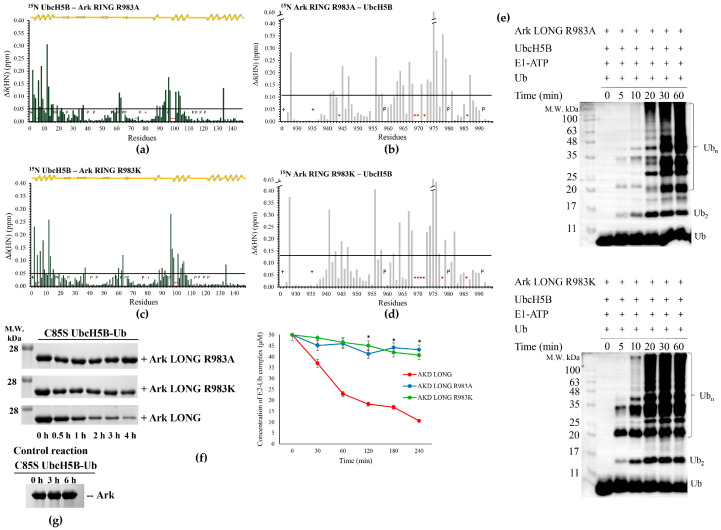
The role of R983 in Arkadia-mediated ubiquitylation. (**a**) Diagram of the total CSPs measured at a 1:2 molar ratio for ^15^N UbcH5B/^14^N Ark RING R983A, *: represents disappeared residues, +: represents residues with no information. Diagram of the total CSPs measured at a 1:2 molar ratio for (**b**) ^15^N Ark RING R983A/^14^N UbcH5B, (**c**) ^15^N UbcH5B/^14^N Ark RING R983K, and (**d**) ^15^N Ark RING R983K/^14^N UbcH5B. (**e**) (**Top**), in vitro auto-ubiquitylation assays of Ark LONG R983A and (**bottom**), in vitro ubiquitylation assays of Ark LONG R983K. (**f**) (**Left**), oxyester hydrolysis assays showing the disappearance of C85S UbcH5B-Ub in the presence of Ark LONG, Ark LONG R983A, and Ark LONG R983K over time. (**Right**), densitometry quantification from gels (bars indicate the range of experimental duplicates, * indicates statistically significant difference, *p* value ≤ 0.05). (**g**) Control reaction demonstrating the stability of oxyester complex over time. The uncropped WB images can be seen in Appendix A.

**Figure 6 cancers-15-01040-f006:**
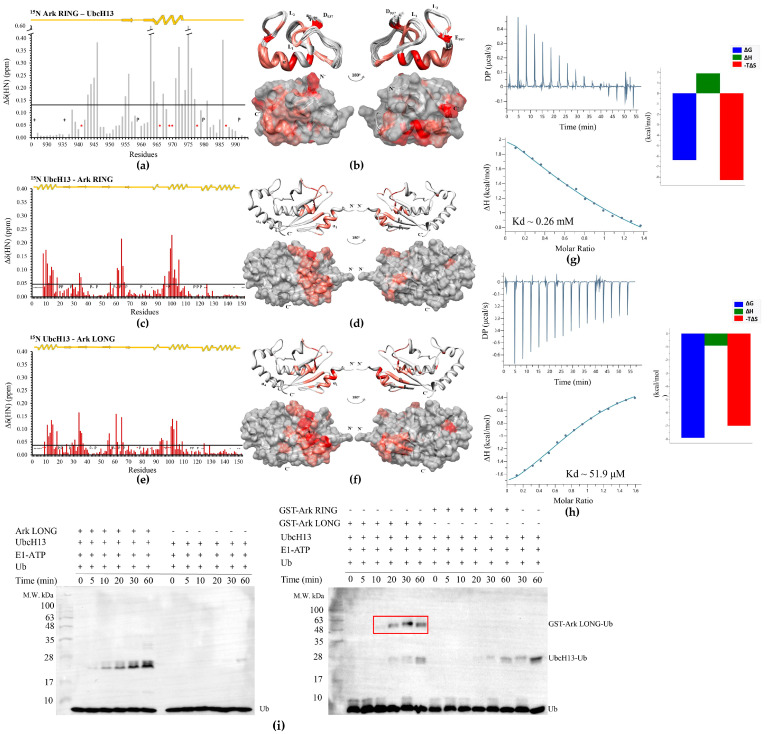
Interaction of E3 Arkadia with UbcH13 E2 enzyme. (**a**) Diagram of the total CSPs measured at a 1:2 molar ratio (saturation point) for ^15^N Ark RING/^14^N UbcH13, *: represents disappeared residues, +: represents residues with no information. (**b**) Ark RING mapping (PDBid: 2KIZ) after the addition of UbcH13 (residues that disappeared during the interaction are colored red, whereas residues that exhibited ‘fast exchange’ interaction are colored coral). (**c**) Diagram of the total CSPs measured at a 1:2 molar ratio for ^15^N UbcH13/^14^N Ark RING. (**d**) UbcH13 mapping (PDBid: 1J7D) after the addition of Ark RING. (**e**) Diagram of the total CSPs measured at a 1:2 molar ratio for ^15^N UbcH13/^14^N Ark LONG. (**f**) UbcH13 mapping (PDBid: 1J7D) after the addition of Ark RING. (**g**) ITC data for the titration of UbcH13 into Ark RING and (**h**) into Ark LONG. (**i**) In vitro auto-ubiquitylation assays using UbcH13 in the presence and absence of Ark LONG (**left**) and in the presence of GST-Ark LONG or GST-Ark RING (**right**).

**Figure 7 cancers-15-01040-f007:**
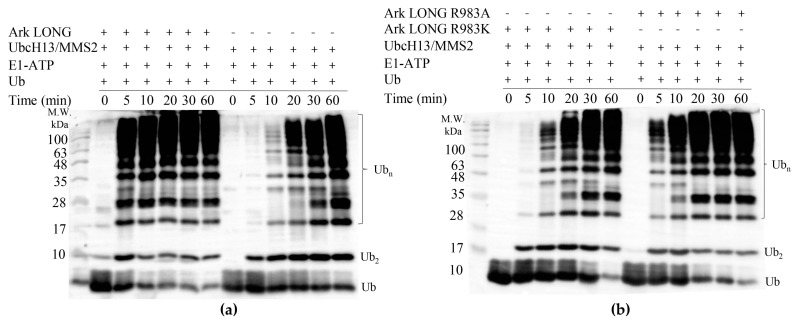
(**a**) In vitro ubiquitylation assays using UbcH13/MMS2 complex in the presence and absence of Ark LONG. (**b**) In vitro ubiquitylation assays using UbcH13/MMS2 complex in the presence of Ark LONG R983A or R983K.

**Figure 8 cancers-15-01040-f008:**
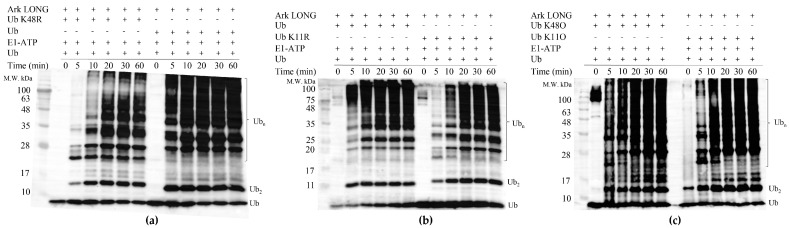
In vitro ubiquitylation assays using UbcH5B in the presence of (**a**) Ub K48R and wt Ub, (**b**) Ub K11R and wt Ub, and (**c**) Ub K48O and Ub K11O.

**Figure 9 cancers-15-01040-f009:**
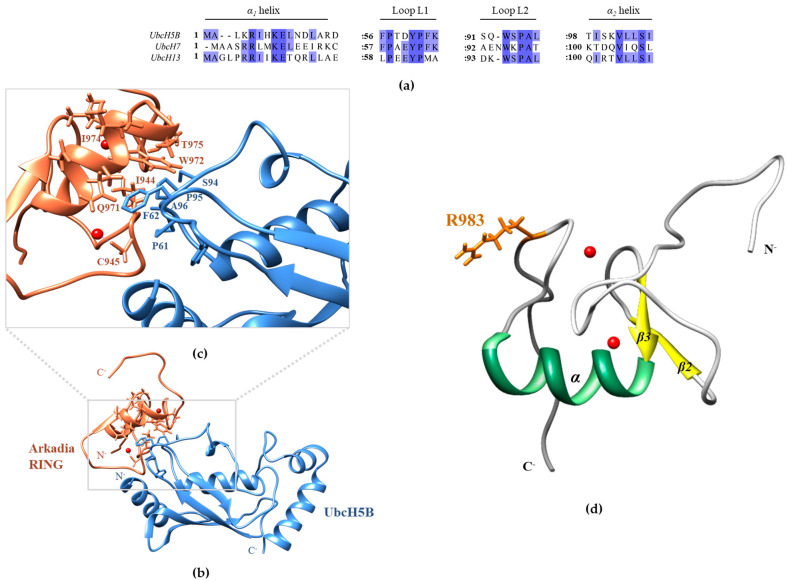
(**a**) Sequence alignment of helix *a_1_*, loop L1, loop L2, and helix *a_2_* of UbcH5B, UbcH7, and UbcH13 E2 enzymes, showing the amino acid conservation among the interacting with Arkadia regions. (**b**) Model of the UbcH5B–Ark RING complex (UbcH5B in light blue and Ark RING in coral). The model was prepared by superimposing Ark RING (PDBid: 2KIZ) to the X-ray structure of the UbcH5B-c-CBL complex (PDBid: 4A49) and extracting the c-CBL structure. Zn(II) ions are depicted as red spheres. (**c**) Close-up of the theoretical UbcH5B–Ark RING interface highlighting the central role of UbcH5B’s Phe62 and Ser94-Pro95-Ala96 motif. Key contact residues from Arkadia and UbcH5B are shown as coral and light blue sticks, respectively. (**d**) NMR structure of Ark RING (PDBid: 2KIZ) with Arg983 shown in orange sticks.

**Table 1 cancers-15-01040-t001:** Thermodynamic parameters of the Arkadia constructs and mutants’ interaction with the E2 enzymes and their mutants obtained by ITC at 25 °C.

Protein	Titrant	*K*_d_ (μM)	N	ΔG (Kcal/mol)	ΔH (Kcal/mol)	−TΔS (Kcal/mol)
Ark RING	UbcH5B	32 ± 2	1	−6.1	3.1	−9.3
Ark LONG	UbcH5B	2.7 ± 0.5	0.6	−7.6	−4.7	−2.9
Ark LONG R983A	UbcH5B	8.4 ± 0.4	0.7	−6.9	−5.8	−1.1
Ark LONG R983K	UbcH5B	5.5 ± 0.6	0.5	−7.2	−3.6	−3.6
Ark RING	UbcH5B F62A	130 ± 17	ND	−5.3	3.6	−8.9
Ark LONG	UbcH5B F62A	194 ± 24	ND	−5.0	2.9	−7.9
Ark RING	UbcH7	27.3 ± 2	0.9	−6.2	2.2	−8.4
Ark LONG	UbcH7	4.3 ± 0.5	0.7	−7.3	−0.7	−6.5
Ark RING	UbcH7 K96S	10.2 ± 0.6	1	−6.8	−0.9	−5.8
Ark RING	UbcH13	264 ± 46	1.1	−4.8	3.4	−8.3
Ark LONG	UbcH13	51.9 ± 5	0.9	−5.8	−2.3	−3.5

Each value is the average of at least three independent measurements. Abbreviations: N, stoichiometry; *K*_d_, dissociation constant; ΔH, ΔS, and ΔG, changes in binding enthalpy, binding entropy, and Gibbs energy, respectively.

## Data Availability

Not applicable.

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
