# Peer review of "E2 Partner Tunes the Ubiquitylation Specificity of Arkadia E3 Ubiquitin Ligase"

_cancers, 2023, doi:10.3390/cancers15041040_

Round 1

Reviewer 1 Report

In this manuscript the interaction of Arkadia with E2 partners UbcH5B, UbcH13 and UbcH7 has been investigated by NMR spectroscopy and ITC. These study showed that the E2-Arkadia interaction surface is similar in all pairs examined, although the interaction affinity differs significantly depending on the specific E2 enzyme. In this study, the authors also investigated the role of some of the residues of E2 partners involved in the interaction with Archadia in impacting in the protein-protein interaction/affinity and in the ubiquitination machinery. Overall, this work is well-performed and the conclusions are consistent with the reported experimental data. There are only some aspects that needs to be modified or clarified:

1.      How assignment of Ala 62 mutant of UbcH5B has been achieved? Selective 15N labeling of UbcH5B F62A with 15N-Ala is indeed not sufficient to obtain it. The assignment needs to be submitted to BMRB.

2.      The data of wt UbcH5B are not reported in Fig 2 as mentioned by the authors in this sentence “Further CSP analysis of the UbcH5B F62A residues after addition of Ark RING (Fig. 2c) showed weaker interaction compared to the titration with the wt UbcH5B

3.      In fig2 the mapping of CSP on the structure of UbcH5B needs to be added.

4.      The authors mentioned that: “E2 enzyme UbcH7 which has 37% percentage identity to the UbcH5B enzyme but lacks the SPA motif” but this is not correct as Fig 1 showed that UbcH7 just modified the Ser with Lys while Pro and Ala are not altered. So the study on UbcH7 just monitor a single mutation effect and not the role of SPA segment. The data on UbcH7 are thus not well reported in the manuscript as SPA motif was not absent in UbcH7 but just a Set->Lys mutant was investigated.

5.       This data “However, despite the noteworthy enhancement in E2-E3 affinity attributable to K96S mutation, the mutant failed to form an enzymatically active E2-E3 pair with Ark LONG and promote auto-ubiquitination (Fig. 3h).” need to be rationalized.

6.      The procedure of the assignment of the HN resonances of the S22R mutant is not reported. Reverse labeling of 15N UbcH5B S22R performed for arginines is not sufficient to obtain resonance assignment. The assignment needs to be submitted to BMRB. The same comment holds for the other two mutants, R983A Ark RING and K96S UbcH7.

Reviewer 2 Report

This manuscript by Delegkou et al. focuses on the biophysical examination of the RING E3 ubiquitin ligase Arkadia (aka RNF111) and its interactions with various E2 ubiquitin conjugating enzymes (i.e. UbcH5b, UbcH13, UbcH7).  Specifically, the authors employed multidimensional NMR spectroscopy, ITC, and auto-ubiquitylaton assays to assess the importance of various residues at the interface of the Arkadia/E2 complexes they studies.  Using 15N-enriched proteins and measuring chemical shift perturbation with increasing concentrations of unlabeled proteins, the authors were able to identify subtle differences on the surface of Arkadia and the E2s examined.  Their ITC experiments also demonstrate significant differences in binding affinity for each Arkadia RING/E2 pair – but which E2 does Arkadia prefer?  The authors also identified and verified the importance of a conserved Phe and “SPA” motif in the class I E2s they examined using ITC and auto-ubiquitylation assays.  The manuscript reads well with strong data and a compelling discussion of the literature.  However, suggestions are listed below that could help to strengthen their paper and findings.

1.     The lack of auto-ubiquitylation activity with UbcH7 is not surprising as this E2 exclusively discharges its ubiquitin cargo on to Cys residues (typically to the catalytic Cys of an RBR or HECT E3), not directly to a Lys (please see Wenzel et al. 2011 Nature).

2.     The differences observed for with UbcH5B wt and S22R oxyester complex breakdown could be related to higher order complexes being formed that make the oxyester in UbcH5B being more easily hydrolyzed (please see Sakata et al. 2015 Structure).  While this is alluded to in the discussion, it needs to be discussed earlier in the results section too.

3.     The data presented in figure 8 suggests that only K48 and K11 polyubiquitin chains are generally overexposed making it difficult to completely delineate chain specificity.  AQUA-MS to assess the specific ubiquitin chain linkages is recommended.

4.     Analogous data RE: the importance of W972 in the Arkadia RING domain, this was also previously reported with Rbx1 and its interaction with CDC34 (UBE2R1; please see Spratt et al. 2012 JBC).

5.     It is strongly encouraged that the authors perform their in vitro assays in the presence of a bona fide Arkadia substrate (perhaps SMAD7?) to verify that the UbcH5B-dependent activity they are observing correlates with the reported Arkadia facilitated ubiquitylation activity that happens in the cell.

Minor suggestions:

Since the attachment of ubiquitin to a protein is a posttranslational modification, the correct term that should be used throughout this review is “ubiquitylation” and "ubiquitylate".  This is analogous to phosphorylation, methylation, acetylation, acylation, etc.

All acronyms need to be defined as soon as they are introduced in the text.  Some are, but many are not.

Author Response

Response to Reviewer 2 Comments

Point 1: The lack of auto-ubiquitylation activity with UbcH7 is not surprising as this E2 exclusively discharges its ubiquitin cargo on to Cys residues (typically to the catalytic Cys of an RBR or HECT E3), not directly to a Lys (please see Wenzel et al. 2011 Nature).

Response 1: UbcH7 indeed functions with HECT and RBR ligases and requires transthiolation reaction to transfer Ub to an E3 ligase. The submitted manuscript has been revised according to reviewer’s comments at rows 363-369.

Point 2: The differences observed for with UbcH5B wt and S22R oxyester complex breakdown could be related to higher order complexes being formed that make the oxyester in UbcH5B being more easily hydrolyzed (please see Sakata et al. 2015 Structure). While this is alluded to in the discussion, it needs to be discussed earlier in the results section too.

Response 2: The submitted manuscript has been revised according to reviewer’s comments at rows 400-404.

Point 3: The data presented in figure 8 suggests that only K48 and K11 polyubiquitin chains are generally overexposed making it difficult to completely delineate chain specificity. AQUA-MS to assess the specific ubiquitin chain linkages is recommended.

Response 3: It is indeed a very good suggestion. We aim to assess the specific ubiquitin chain linkages through AQUA-MS and SEC-MALS in our future work.

Point 4: Analogous data RE: the importance of W972 in the Arkadia RING domain, this was also previously reported with Rbx1 and its interaction with CDC34 (UBE2R1; please see Spratt et al. 2012 JBC).

Response 4: The manuscript has been corrected according to the reviewer’s suggestion.

Point 5: It is strongly encouraged that the authors perform their in vitro assays in the presence of a bona fide Arkadia substrate (perhaps SMAD7?) to verify that the UbcH5B-dependent activity they are observing correlates with the reported Arkadia facilitated ubiquitylation activity that happens in the cell.

Response 5: We thank the reviewer for this insightful comment. However, the aim of the present study is to elucidate in vitro the structural, biochemical and functional properties of Arkadia in cooperation with different E2 enzymes and provide valuable, novel, atomic and molecular level insights on how Arkadia recognizes its physiological E2 partners and leads to diverse linkage specificity of the poly-ubiquitin chain, rather than correlating in vitro with in vivo function. Hopefully these findings will improve our knowledge on the Arkadia-mediated ubiquitylation that takes place in the cell, especially the ubiquitylation of the substrates of Arkadia.

Minor suggestions:

Point 6: Since the attachment of ubiquitin to a protein is a posttranslational modification, the correct term that should be used throughout this review is “ubiquitylation” and "ubiquitylate". This is analogous to phosphorylation, methylation, acetylation, acylation, etc.

Response 6: Although the term “ubiquitination” is widely used in the literature, the manuscript has been thoroughly revised accordingly.

Point 7: All acronyms need to be defined as soon as they are introduced in the text. Some are, but many are not.

Response 7: Τhe manuscript has been revised accordingly.
